# Scale for students' attitude towards AIGC feedback in english pronunciation learning: Development, validation and application

Weihe Zhong[1], Yanchao Yang[2,3]*, Bosheng Jing[4], Xinxin Yang[5], Zehan Tan[6], Qiu Wei[1,7]

1 Rector's Office, Macau Millennium College, Macao, China, 2 Institute of International Language Services Studies, Macau Millennium College, Macao, China, 3 Qinggong College, North China University of Science and Technology, Tangshan, Hebei Province, China, 4 School of Humanities and Languages, The University of New South Wales, Sydney, Australia, 5 Faculty of Humanities and Social Sciences, Macau Millennium College, Macao, China, 6 Faculty of Digital Science and Technology, Macau Millennium College, Macao, China, 7 Faculty of Health and Wellness, City University of Macau, Macao, China

* yangyanchao@mmc.edu.mo

## Abstract

This study develops and validates the Scale of Students' Perception of AIGC Feedback for English Pronunciation Learning. The research was conducted at a university in northern China using a convenience sampling method. The exploratory factor analysis (EFA) involved 207 participants, while the confirmatory factor analysis (CFA) included 229 participants. Based on interviews with 10 students who had used AIGC tools for English pronunciation learning, 16 representative items were identified. Expert validation was performed through interviews with 8 experts—four English pronunciation teachers with extensive experience using AIGC in teaching, and four AIGC specialists. Content validity was confirmed, and all items were retained. The EFA results revealed four dimensions: Accuracy, Strictness, Clarity, and Personalisation. The CFA results demonstrated good structural and convergent validity. However, the discriminant validity was slightly problematic. Concurrent validity was confirmed by the high correlation between the scale and perceived English Pronunciation Self-efficacy. The study has several limitations, including its cross-sectional design, limited sample diversity, and reliance on traditional validation methods (EFA and CFA), suggesting the need for test-retest reliability, a more diverse sample, and alternative methods like Item Response Theory (IRT) or Network Analysis in future research. The validated scale offers valuable insights into how students perceive and interact with generative AI tools, and it can serve as a useful instrument for educators and researchers interested in exploring the impact of AI feedback systems on language learning.

**Data availability statement:** All relevant data are within the Supporting Information files.

**Funding:** The author(s) received no specific funding for this work.

**Competing interests:** The authors have declared that no competing interests exist.

## Introduction

The emergence of generative artificial intelligence (AIGC) has reshaped the landscape of technological advancements, bringing about a significant transformation across various industries, such as in healthcare [1,2], hospitality and tourism industry [3,4], commerce [5,6], and more notably, education [7–9]. In the context of education, AIGC presents an opportunity to reimagine traditional learning processes. Its application goes beyond automating tasks to providing personalized and adaptive learning experiences that respond to individual student needs. As such, AIGC is poised to become a foundational tool for enhancing teaching effectiveness [10,11], improving student engagement [12,13], and fostering a more dynamic and efficient educational environment.

Among the various domains in which AIGC is being deployed, the field of education is one of the most promising, particularly in the context of language learning (e.g., [14–16]). The integration of AIGC technologies in language education offers a variety of specific functions that address the challenges faced by language learners. These technologies provide personalized learning paths, adaptive content creation, and real-time feedback. For instance, intelligent tutoring systems can analyze learners' performance and tailor lessons to their proficiency level, ensuring targeted skill development [17,18]. Additionally, AIGC-powered platforms can generate customized exercises and quizzes, enabling learners to practice specific language areas where they need improvement, such as vocabulary building or grammar usage [19]. Furthermore, AIGC tools can offer instant translations, helping learners bridge language gaps [20], while automated writing assistants can help refine written expression through grammar correction and style suggestions [21,22]. These capabilities have significantly transformed the traditional language learning model by offering more interactive, individualized, and efficient learning experiences.

Specifically, pronunciation stands as a fundamental component that significantly influences students' communicative competence [23–25]. Effective pronunciation not only enhances the clarity of speech but also contributes to the overall success of language learners in real-life interactions. Despite its importance, pronunciation instruction in traditional classroom settings often faces challenges, such as inadequate exposure to the language as it is spoken in the real world [26], limited resources [27], influence of mother tongue [28]. This is where AIGC tools can offer substantial value. By leveraging speech recognition [29,30], speech synthesis [31], Natural Language Processing [32] and machine learning algorithms [33], AIGC technologies are capable of providing precise and immediate feedback on pronunciation, offering students the opportunity to refine their speech patterns autonomously. Such tools allow for a more interactive and efficient learning process, empowering students to engage with the material more effectively and enabling educators to focus on other aspects of language instruction [34–37].

The effectiveness of AIGC tools in English pronunciation learning is intrinsically linked to students' attitudes toward the feedback they receive. Students' attitudes toward the accuracy, clarity, and relevance of the feedback provided by AIGC tools

play a critical role in determining how they engage with these technologies and how effectively they benefit from them. If students view the feedback as accurate, personalized, and conducive to their learning, they are more likely to integrate these tools into their study routines, leading to improved pronunciation skills. On the other hand, if the feedback is perceived as unclear, overly strict, or not personalized to individual needs, students may disengage from the learning process, diminishing the potential benefits of AIGC tools. However, despite the significant amount of research on attitudes toward generative artificial intelligence technology and products [38–44], there is limited research on attitudes toward generative artificial intelligence's feedback on English pronunciation. Therefore, the primary objective of this study is to investigate university students' attitudes toward AIGC feedback in the context of English pronunciation learning by developing and validating a scale for measuring these attitudes, with the aim of facilitating more effective feedback mechanisms.

## Literature review

### Feedback

Feedback plays a vital role in the teaching-learning process, acting as a bridge between students' current understanding and their desired learning outcomes. It not only helps students identify gaps in their knowledge but also provides them with clear guidance on how to improve [45]. Through feedback, students are given the chance to address and improve upon areas where their knowledge or skills may be lacking. This allows them to focus on specific weaknesses, whether they involve gaps in understanding, incorrect concepts, or underdeveloped abilities [46].

Feedback can come from both human and technological sources, each offering unique advantages. Human sources, such as teachers [47–50] and peers [51–53], provide personalized, context-specific feedback that can address individual needs, clarify misunderstandings, and offer emotional support. Teachers, in particular, offer expert insights that guide students through the learning process. Peer feedback, on the other hand, encourages collaborative learning, promotes critical thinking, and provides students with different perspectives on their work. In contrast, technological tools bring a new dimension to feedback [54,55]. These systems can provide immediate, data-driven feedback, offering insights on language use, content accuracy, and even structure. With advancements in technology, generative artificial intelligence (AI) feedback has now emerged as a powerful tool in the learning process. Unlike traditional systems, generative AI can not only analyze and assess content but also generate customized feedback that adapts to individual students' needs [56–58]. This feedback is often immediate and highly personalized, allowing students to receive tailored suggestions and corrections in real time, promoting more efficient and effective learning. As AI continues to evolve, its ability to provide nuanced, context-specific feedback is expected to further enhance the learning experience across various disciplines.

Studies have consistently shown that feedback has a beneficial effect across a wide range of disciplines. For instance, in language learning context [59–62], feedback helps improve vocabulary acquisition [63–65] and speaking proficiency as well as skills [66–68].

### Attitude

The attitude construct has been recognized as one of the most indispensable concepts in psychology. Over time, it has been defined in various ways, reflecting its complexity and relevance across different contexts. For instance, it was viewed as a psychological, evaluative response toward a specific person, place, thing, event, or other object, characterized by positive and/or negative feelings, and shaped by affective, behavioral, and cognitive information [69–71].Additionally, it was also characterized as a stable and broad assessment of an object, person, group, issue, or concept, evaluated on a spectrum from negative to positive, offers overall assessments of target objects and are frequently thought to stem from particular beliefs, emotions, and previous behaviors linked to those objects. [72]. Furthermore. it can also be described as an individual's perspective and evaluation of something or someone, reflecting a tendency or inclination to respond either positively or negatively to a particular idea, object, person, or situation [73]. Additionally, an attitude is seen as a stable and lasting evaluation or emotional response to a stimulus, object, or situation, which can be either positive or negative.

This evaluation plays a key role in shaping the behaviors directed toward the attitude object [74]. In conclusion, the central idea shared across the various definitions of attitude is that it involves an individual's evaluation or assessment of something, whether it be a person, object, event, or situation.

Traditionally, attitude is organized into three dimensions: Cognitive (which involves perceptions and beliefs), Affective (which includes likes, dislikes, feelings, or emotions), and Behavioral (which refers to actions or intentions toward the object, influenced by the cognitive and affective responses) [75]. Empirical research, however, does not provide clear evidence distinguishing the thoughts, emotions, and behavioral intentions linked to a specific attitude [69,76]. The model does not fully explain the interplay between the affective, cognitive, and behavioral components, which often intertwine in real-world situations. In practice, these dimensions are not isolated from one another; instead, they continuously influence and shape each other in dynamic ways. The linear and distinct framework of the ABC model, therefore, may not fully account for the complexities of how attitudes manifest in behavior, particularly when emotional and cognitive responses are deeply interlinked and context-dependent. Consequently, a more nuanced approach may be needed to capture the fluid nature of these dimensions and their influence on each other.

## Methods

### Ethical considerations

The study was approved by the Ethics Committee (Approval No: MMCIRB-2024–002), ensuring compliance with ethical standards. Data collected was anonymized, with no identifiable information retained, and strict measures were implemented to protect participants' privacy. A convenience sampling method was used to collect data at an independent college in northern China, where students from various provinces in China, representing diverse cultural, educational, and socio-economic backgrounds, participated, making the sample reasonably representative. The survey link created by Wenjuanxing was distributed via WeChat. The consent was obtained in electronic format: participants were required to read the informed consent form embedded in the online questionnaire, and by clicking the "Agree to participate" option, they indicated their consent to participate. Only after providing consent in this way could they proceed to complete the survey. Furthermore, all data was securely stored and used solely for this research, with strict confidentiality measures in place to safeguard participants' privacy.

### Participants

A convenience sampling method was employed to collect data from students at an independent college in northern China. Participants completed an online questionnaire via Wenjuanxing on 20th December 2024, with an average completion time of around 3 minutes. The effective sample size was 436, based on a criterion of 2 seconds per item for valid responses [77,78]. The participants had a mean age of 18.88 years. The sample was then randomly divided into two groups for further analysis: 207 participants were assigned to the exploratory factor analysis (EFA) group (see S1 Table in Supporting Information for reference), and 229 participants were allocated to the confirmatory factor analysis (CFA) group (see S2 Table in Supporting Information for reference).

The demographic characteristics of the sample in both the EFA and CFA groups are displayed in Table 1: In terms of sex, the EFA group consisted of 36.7% male participants (n = 76) and 63.3% female participants (n = 131). In the CFA group, 41.5% of participants were male (n = 95), and 58.5% were female (n = 134). Regarding birthplace, 23.2% of participants in the EFA group were from urban areas (n = 48), while 76.8% were from rural areas (n = 159). In the CFA group, 19.7% of participants were from urban areas (n = 45), and 80.3% were from rural areas (n = 184). These demographic distributions provide a broad representation across both sex and geographical background in the sample.

### Scale for students' attitude towards AIGC feedback in english pronunciation learning

The Scale for Students' Attitude towards AIGC Feedback in English Pronunciation Learning was developed through a rigorous and systematic process to ensure the scale's accuracy, relevance, and clarity.

**Table 1. Demographic information.**

| Demographic variable | Group | EFA Groups | | CFA Groups | |
|---|---|---|---|---|---|
| | | Frequency | Percent (%) | Frequency | Percent (%) |
| Sex | Male | 76 | 36.715 | 95 | 41.485 |
| | Female | 131 | 63.285 | 134 | 58.515 |
| Birthplace | Urban | 48 | 23.188 | 45 | 19.651 |
| | Rural | 159 | 76.812 | 184 | 80.349 |
| Total | | 207 | 100 | 229 | 100 |

The items for the scale were generated through focus group interview with 10 students who had experience using generative artificial intelligence to assist in their English pronunciation learning. Ten students, all recommended by their English teachers, participated in the interviews. These students were recommended based on their prior experience with using generative artificial intelligence tools to assist in their English pronunciation learning during their regular study sessions.

The interviews explored students' experiences and perceptions regarding the feedback they received from AIGC systems. Sample questions included: *Do you think the generative AI software can accurately identify your pronunciation errors? If you repeatedly make the same pronunciation mistakes, how does the generative AI software handle these errors? Is the feedback provided by the generative AI software easy to understand? Does the generative AI software remember your previous pronunciation issues and provide more targeted suggestions in subsequent feedback?* The focus group interview lasted approximately 40 minutes, which allows for rich, interactive discussions, where participants can build upon each other's responses, creating a dynamic exchange of ideas and help the researchers identify key themes and areas of interest in the feedback process.

The interview data were analyzed using a Grounded Theory methodology, which allowed for a detailed and systematic exploration of the students' responses. In the first phase of the analysis, open coding was conducted, where the researchers carefully examined each interview transcript line by line, identifying discrete themes, phrases, and concepts that emerged from the data. These initial codes were assigned to specific segments of the data to capture the essence of the responses.

For instance, some responses were coded with labels such as "phoneme errors," "vowel mispronunciation," and "stress errors" Additionally, some responses included terms like "repeated corrections," "zero tolerance for errors," and "attention to minor mistakes." Furthermore, some responses were coded with labels such as "clear instructions," "simple language," and "avoid technical jargon". Moreover, some responses were coded with labels such as "customized exercises," "tracking progress," and "targeted feedback". These labels helped capture the specific aspects of pronunciation that the AI was addressing.

During the axial coding phase, the researchers reviewed the open codes and began to group related themes and concepts into broader categories, focusing on the relationships between them. This step helped to refine the themes and identify patterns, such as how feedback on pronunciation accuracy, clarity, and personalization were perceived by students.

The final categories were structured into coherent groupings that captured the key aspects of students' experiences and perceptions, providing a foundation for the theory-building process. This coding process led to the identification of four primary dimensions that reflected the students' attitudes towards the AIGC feedback: Accuracy, Strictness, Clarity, and Personalization.

Once the items were developed, they were reviewed by 8 English language teachers who were experienced in AI-assisted language learning. These experts were asked to evaluate the items and proposed categories based on their clarity,

representativeness, and relevance. To assess the content validity of the scale, the experts rated each item on a 4-point scale, allowing for the calculation of the Item-Content Validity Index (I-CVI) and Scale-Content Validity Index (S-CVI). Both indices were calculated to determine the extent to which the items represented the core concepts of the scale. The results from the expert review indicated that the I-CVI and S-CVI scores met the acceptable threshold, confirming the content validity of the scale. No items required revision based on the feedback from the experts. This step ensured that the scale accurately reflected the key dimensions of students' attitudes towards AI-generated feedback. To measure students' attitudes, a 5-point Likert scale was used, with higher scores indicating stronger agreement with the statements provided.

### English pronunciation self-efficacy

To validate the concurrent validity of the Scale for Students' Attitude towards AIGC Feedback in English Pronunciation Learning, the English Pronunciation Self-Efficacy questionnaire was also collected. This additional data helps assess the relationship between students' attitudes towards AIGC feedback and their self-perceived efficacy in English pronunciation, ensuring that the scale aligns with relevant established measures in the context of English pronunciation learning. The English Pronunciation Self-Efficacy scale [79] is designed to measure students' self-efficacy in English pronunciation, consisting of two dimensions: Segmental Features (4 items) and Suprasegmental Features (5 items). The scale has been validated using both Rasch and Classical Test Theory (CTT) methods, with excellent reliability and validity results. Furthermore, the scale demonstrates strong generalizability across variables such as gender, major, student domicile, and time. The scale uses a five-point Likert scale, with higher scores indicating greater self-efficacy in English pronunciation.

### Analytical procedure

The scale development process involved several steps to ensure its reliability and validity. First, an item analysis was conducted, including independent samples t-tests, item-total correlation analysis, and Cronbach's α if item deleted. Next, factor analysis (EFA and CFA) was performed to identify and confirm the scale's underlying factor structure. During Confirmatory Factor Analysis (CFA), model fit was assessed using fit indices such as RMSEA, SRMR, CFI, TLI, and chi-square statistics, with values of RMSEA and SRMR below 0.08, and CFI and TLI above 0.90 considered indicative of a well-fitting model [80,81]. Convergent validity was assessed using factor loadings, composite reliability, and average variance extracted (AVE), while discriminant validity was evaluated using the Fornell-Larcker criterion. Finally, the concurrent validity was conducted by calculating the Pearson correlation with English Pronunciation Self-Efficacy scale. This rigorous approach ensured that the scale accurately measured the intended construct and demonstrated both reliability and validity.

## Results

### Item analysis

The item discrimination analysis, based on a t-test between the top 27% (total ≥ 58) and bottom 27% (total ≥ 48) groups, revealed significant differences for all items ($p = 0.01$), confirming their ability to differentiate effectively [82]. Item-total correlations ranged from 0.811 to 0.867 ($p = 0.01$), demonstrating strong relationships with the overall construct and supporting internal consistency. The overall Cronbach's α was 0.972, indicating excellent reliability. Furthermore, analysis showed that removing any item did not improve Cronbach's α, affirming that all items meaningfully contribute to the scale's consistency.

### Exploratory factor analysis

The results of the exploratory factor analysis (EFA) conducted using the first dataset demonstrated strong evidence for the appropriateness of the data for factor extraction. Specifically, the Kaiser-Meyer-Olkin (KMO) measure was

0.954, indicating excellent sampling adequacy, while Bartlett's Test of Sphericity yielded a significant chi-square value ($\chi^2 = 3518.736$, df = 120, $p < 0.001$), confirming the suitability of the data for factor analysis. Based on the hypothesized factor structure, with the number of factors fixed at four, promax oblique rotation was employed, given the potential correlations among the factors. Items were retained based on the following criteria: (a) factor loadings below 0.40, (b) communalities below 0.30, (c) cross-loadings of items (loadings above 0.30 on two or more factors), and (d) factors with fewer than three items. No items were removed, and the final factor structure was in alignment with the qualitative analysis results as shown in Table 2.

The four factors identified in the scale were named as Personalisation, Accuracy, Strictness, and Clarity, each representing a key dimension for assessing students' attitudes towards the feedback provided by generative AI tools in English pronunciation learning. Personalisation reflects how well the GenAI tailors feedback based on the learner's individual history and pronunciation issues. Accuracy represents the GenAI tool's precision in identifying and correcting specific pronunciation errors, such as phonemes, stress, and intonation. Strictness captures the GenAI's approach in addressing all errors, ensuring they are corrected until fully resolved. Lastly, Clarity pertains to how easily students can comprehend and act upon the feedback given. As shown in Table 2 and 3, Personalisation (Cronbach's $\alpha = 0.927$) explained 21.4% of the variance; Accuracy (Cronbach's $\alpha = 0.940$) explained 20.4% of the variance; Strictness (Cronbach's $\alpha = 0.926$) explained 18.4% of the variance; and Clarity (Cronbach's $\alpha = 0.925$) explained 17.6% of the variance. These factors demonstrated strong internal consistency and accounted for a substantial portion of the total variance.

## Confirmatory factor analysis

CFA was subsequently performed by using the CB-SEM module in SmartPLS to validate the four-dimensional factorial structure of the scale, as illustrated in Fig 1. The model fit was evaluated based on several commonly used fit indices, including Root Mean Square Error of Approximation (RMSEA), Standardized Root Mean Square Residual (SRMR), Comparative Fit Index (CFI), and Tucker-Lewis Index (TLI). According to established criteria, RMSEA values below 0.08, SRMR values below 0.08, and CFI and TLI values above 0.90 indicate an adequate model fit (Hu & Bentler, 1999; McDonald &

**Table 2. Factor loadings.**

| Dimension | Item | Factor 1 | Factor 2 | Factor 3 | Factor 4 | Uniqueness | Commonality | Cronbach's alpha |
|---|---|---|---|---|---|---|---|---|
| Accuracy | A1 | | 0.844 | | | 0.215 | 0.785 | 0.940 |
| | A2 | | 0.724 | | | 0.217 | 0.783 | |
| | A3 | | 0.830 | | | 0.186 | 0.814 | |
| | A4 | | 0.821 | | | 0.148 | 0.852 | |
| Strictness | S1 | | | 0.780 | | 0.232 | 0.768 | 0.926 |
| | S2 | | | 0.684 | | 0.234 | 0.766 | |
| | S3 | | | 0.829 | | 0.246 | 0.754 | |
| | S4 | | | 0.529 | | 0.214 | 0.786 | |
| Clarity | C1 | | | | 0.814 | 0.203 | 0.797 | 0.925 |
| | C2 | | | | 0.574 | 0.257 | 0.743 | |
| | C3 | | | | 0.906 | 0.177 | 0.823 | |
| | C4 | | | | 0.482 | 0.292 | 0.708 | |
| Personalisation | P1 | 0.639 | | | | 0.249 | 0.751 | 0.927 |
| | P2 | 0.689 | | | | 0.226 | 0.774 | |
| | P3 | 0.644 | | | | 0.269 | 0.731 | |
| | P4 | 0.946 | | | | 0.182 | 0.818 | |

*Note. Applied rotation method is promax.*

**Table 3. Factor characteristics.**

| | Unrotated solution | | | | Rotated solution | | |
|---|---|---|---|---|---|---|---|
| | Eigenvalues | SumSq. Loadings | Proportion var. | Cumulative | SumSq. Loadings | Proportion var. | Cumulative |
| Factor 1 | 11.239 | 11.018 | 0.689 | 0.689 | 3.429 | 0.214 | 0.214 |
| Factor 2 | 0.869 | 0.664 | 0.041 | 0.73 | 3.272 | 0.204 | 0.419 |
| Factor 3 | 0.646 | 0.424 | 0.026 | 0.757 | 2.94 | 0.184 | 0.603 |
| Factor 4 | 0.557 | 0.347 | 0.022 | 0.778 | 2.812 | 0.176 | 0.778 |

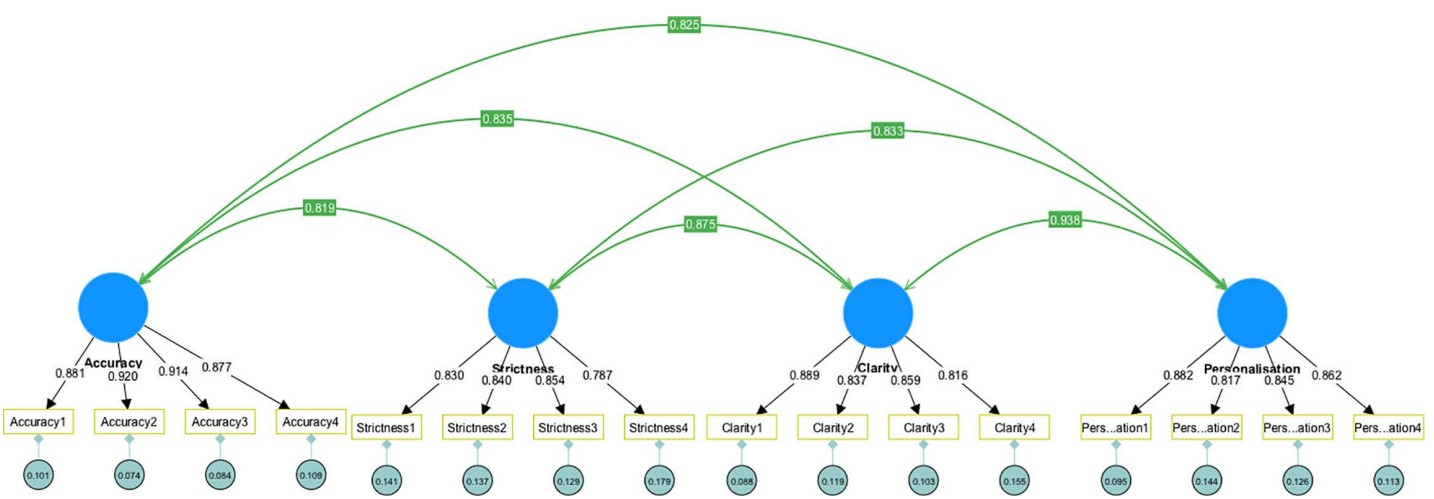

**Fig 1. Four-factor model of scale for students' attitude towards AIGC feedback in English pronunciation learning.**

Ho, 2002). The CFA results presented in Table 4 demonstrated that all fit indices met these thresholds, confirming that the proposed model exhibited an acceptable fit with the data, thus affirming the validity of the underlying factor structure.

## Convergent validity

For convergent validity, both Composite Reliability (CR) and Average Variance Extracted (AVE) were evaluated. As shown in Table 5, the CR values ranged from 0.897 to 0.944, exceeding the recommended threshold of 0.7 [83], indicating that the constructs demonstrate good internal consistency. Similarly, the AVE values ranged from 0.686 to 0.807, surpassing the threshold of 0.5 [83], which confirms that a sufficient proportion of the variance in the indicators is captured by their respective latent variables. These results collectively support the convergent validity of the measurement model.

## Discriminant validity

For discriminant validity, the Heterotrait-Monotrait ratio (HTMT) was used to assess the distinctiveness of the constructs. According to the stricter standard, HTMT values should be below 0.85, while a more lenient threshold allows values up to 0.90 [84]. As displayed in Table 6, the HTMT value for the Personalisation construct was 0.943, which exceeds the 0.90

**Table 4. Model fit results.**

| Chi-square | Degrees of freedom | P value | ChiSqr/df | SRMR | TLI | CFI | RMSEA | RMSEA LOW 90% CI | RMSEA HIGH 90% CI |
|---|---|---|---|---|---|---|---|---|---|
| 240.565 | 98 | 0 | 2.455 | 0.031 | 0.951 | 0.96 | 0.08 | 0.067 | 0.092 |

**Table 5. Convergent validity results.**

|  | Composite reliability (rho_c) | Average variance extracted (AVE) |
| --- | --- | --- |
| Accuracy | 0.944 | 0.807 |
| Clarity | 0.912 | 0.724 |
| Personalisation | 0.913 | 0.726 |
| Strictness | 0.897 | 0.686 |

**Table 6. Discriminant validity results.**

|  | Accuracy | Clarity | Personalisation | Strictness |
| --- | --- | --- | --- | --- |
| Accuracy |  |  |  |  |
| Clarity | 0.836 |  |  |  |
| Personalisation | 0.824 | **0.943** |  |  |
| Strictness | 0.825 | 0.881 | 0.836 |  |

threshold. However, the HTMT values for all other construct pairs remained within the acceptable range, confirming that the constructs are sufficiently distinct for the majority of the model. Thus, while the value for Personalisation is slightly above the threshold, the results partially support the discriminant validity of the measurement model.

## Concurrent validity

For concurrent validity, the English Pronunciation Self-Efficacy scale [79] was used as the benchmark instrument by calculating the Pearson correlation coefficients between English Pronunciation Self-Efficacy and the dimensions as well as the total score of Scale for Students' Attitude towards AIGC Feedback in English Pronunciation Learning. The correlations between the model's dimensions and the English Pronunciation Self-Efficacy were as follows: 0.495 (Accuracy), 0.526 (Strictness), 0.565 (Clarity), and 0.546 (Personalisation) with p < 0.001. The correlation between Scale for Students' Attitude towards AIGC Feedback in English Pronunciation Learning and the English Pronunciation Self-Efficacy scale was 0.581 (p < 0.001). These significant positive correlations suggest that the proposed model is meaningfully related to the English Pronunciation Self-Efficacy scale, supporting its concurrent validity.

## Discussion

The primary aim of this study was to develop and validate the Scale for Students' Attitude towards AIGC Feedback in English Pronunciation Learning. The scale was rigorously developed through a systematic process, including item generation and expert validity checks, to ensure its content relevance and clarity. The validation process involved several robust methods, including item analysis, exploratory factor analysis (EFA), and confirmatory factor analysis (CFA), which together provided a comprehensive assessment of the scale's structure and reliability. Additionally, concurrent validity was assessed through correlation with the English Pronunciation Self-Efficacy scale, further strengthening the scale's validity.

The results of the EFA and CFA both supported a four-factor structure, with the identified dimensions being Accuracy, Clarity, Personalisation, and Strictness. These factors reflect distinct aspects of the students' attitudes toward AIGC feedback in English pronunciation learning. Specifically, the Accuracy dimension evaluates the ability of generative AI tools to precisely identify and correct specific pronunciation errors. It is supported by speech deep learning classification algorithms, which enable the AI to process and recognize phonetic variations, identify incorrect pronunciations, and provide accurate corrections.

The Strictness dimension assesses how rigorously these tools handle errors, ensuring consistent and uncompromising correction until the desired pronunciation standard is achieved. This is achieved through reinforcement learning, where the

GenAI system is trained to reinforce correct pronunciation behaviors and penalize errors, adapting its feedback progressively to maintain a high standard of pronunciation correction.

The Clarity dimension addresses the students' ability to understand and act upon the feedback, highlighting the ease with which students can follow the AI's suggestions. It relies on large language models, which are designed to generate clear, comprehensible, and contextually relevant feedback. These models enhance communication by tailoring responses to the user's level of understanding and language proficiency.

Finally, the Personalisation dimension evaluates the AI's capacity to utilize learning records for providing personalized and continuous improvement feedback tailored to the individual student. This is supported by database technology, which store and analyze students' learning histories and performance data, allowing the GenAI to adjust feedback to meet each student's unique needs and track their progress over time.

The scale demonstrated high reliability across its dimensions, further confirming its suitability for assessing students' attitudes toward GenAI-generated feedback in pronunciation learning.

The scale demonstrates strong convergent validity and concurrent validity, which support its overall reliability and relevance in measuring students' perceptions of AIGC feedback for English pronunciation learning. However, the discriminant validity presents a slight concern. The HTMT results suggest that the Personalisation construct may not be sufficiently distinct from other constructs in the model, particularly Clarity or other dimensions related to the feedback mechanism. The higher HTMT value for Personalisation (0.943) exceeds the more lenient threshold of 0.90, indicating that it shares significant overlap with other constructs, which may compromise its distinctiveness. The higher HTMT value for Personalisation may stem from its inherent overlap with other constructs in the model, such as Clarity, which could also involve aspects of individualized feedback or customization. Given that Personalisation refers to the degree to which feedback is tailored to the individual learner's needs, it is likely that students' perceptions of feedback clarity also influence their perceptions of its personal relevance. This overlap may blur the distinction between the two constructs, leading to the higher HTMT value for Personalisation. Therefore, the slight overlap between Personalisation and other constructs warrants further investigation. Moreover, since all participants were drawn from the same school, similarities in prior instruction, learning environment, and experience may further increase the perceived overlap between these dimensions. As a result, the HTMT value for Personalization is slightly elevated, indicating minor discriminant validity concerns. We caution that this overlap likely reflects the homogeneity of the sample, rather than a flaw in the scale design, and future studies with more diverse samples are recommended to further validate the distinctiveness of these constructs.

## Implications

By developing a scale to measure students' attitudes toward feedback from generative artificial intelligence (AI), the current study contributes to optimizing English pronunciation learning in several ways. First, generative AI can provide personalized, real-time feedback, and students' attitudes and receptiveness to this feedback directly impact their learning outcomes. Therefore, the research offers insights into improving the design of AI tools to better meet students' needs and enhance learning effectiveness. Moreover, this study helps increase the acceptance of AI in language learning by providing valuable feedback on how students perceive generative AI feedback. Understanding their expectations and attitudes allows generative AI developers to refine their tools' design and functionality to better align with students' preferences and habits.

## Limitations and suggestions

The study does have several limitations that should be addressed in future research. First, the current study employs a cross-sectional design, which captures data at a single point in time. This approach limits the ability to assess the stability or consistency of the scale across multiple time points. While the scale demonstrates good internal consistency and construct validity within the current dataset, future research could benefit from incorporating a test-retest reliability measure.

By collecting data at multiple intervals, researchers can assess the scale's stability and ensure that the attitudes captured are consistent over time, thus enhancing the scale's reliability.

Second, while the sample size in this study meets the minimum requirements for conducting factor analyses, there is room for improvement. A larger and more diverse sample would increase the robustness of the results and improve the generalizability of the findings. Moreover, the sample was drawn from a single institution, which limits the ability to extrapolate the results to a broader population. Although various demographic variables were considered, future research should include students from multiple institutions and geographical locations to ensure that the findings are more representative and applicable across different cultural contexts. This will help to strengthen its generalizability of the scale and its applicability to a wider range of learners.

Third, the validation methods employed in this study, namely Exploratory Factor Analysis (EFA) and Confirmatory Factor Analysis (CFA), are traditional and well-established approaches for scale validation. However, these methods have certain limitations, such as their reliance on linearity and fixed factor structures. Future studies could explore alternative, more flexible approaches to validation, such as Item Response Theory (IRT) and Network Analysis. IRT can offer more nuanced insights into how individual items function across different levels of the latent trait (e.g., students' attitudes towards AIGC feedback), while Network Analysis can provide a deeper understanding of the interrelationships between the different dimensions of the scale, highlighting potential dependencies or complexities not captured by traditional factor models.

By addressing these limitations and incorporating these suggestions, future research can further refine the Scale for Students' Attitude towards AIGC Feedback in English Pronunciation Learning and increase its applicability and precision across a broader spectrum of learners and contexts.

## Conclusion

The validated scale offers valuable insights into how students perceive and interact with generative AI tools, and it can serve as a useful instrument for educators and researchers interested in exploring the impact of AI feedback systems on language learning.

## Supporting information

**S1 Table. EFA dataset.**
(XLSX)

**S2 Table. CFA dataset.**
(XLSX)

**S1 Fig. Four-factor model of scale for students' attitude towards AIGC feedback in English pronunciation learning.**
(PBG)

**S2 File. Scale for Students' Attitude towards AIGC Feedback in English Pronunciation Learning.**
(DOCX)

## Author contributions

**Conceptualization:** Weihe Zhong, Yanchao Yang.

**Investigation:** Xinxin Yang.

**Methodology:** Bosheng Jing, Zehan Tan.

**Software:** Yanchao Yang.

**Supervision:** Xinxin Yang.

**Validation:** Yanchao Yang, Bosheng Jing.

**Writing – original draft:** Yanchao Yang.

**Writing – review & editing:** Weihe Zhong, Yanchao Yang, Zehan Tan, Qiu Wei.

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
