## [Decision Letter · Decision Letter 0]

3 Sep 2025

PONE-D-25-26307Scale for Students’ Attitude towards AIGC Feedback in English Pronunciation Learning: Development, Validation and ApplicationPLOS ONE

Dear Dr. Yanchao Yang,

Thank you for submitting your manuscript to PLOS ONE. After careful consideration, we feel that it has merit but does not fully meet PLOS ONE’s publication criteria as it currently stands. Therefore, we invite you to submit a revised version of the manuscript that addresses the points raised during the review process.

We look forward to receiving your revised manuscript.

Kind regards,

Jie Zeng, Ph.D.

Academic Editor

PLOS ONE

2. Please ensure that you have specified a) Did participants provide their written or verbal informed consent to participate in this study?

b) If consent was verbal, please explain i) why written consent was not obtained, ii) how you documented participant consent, and iii) whether the ethics committees/IRB approved this consent procedure."

- In consent please state in Ethics Method section and manuscript if it is written or verbal. If consent was verbal, please explain a) why written consent was not obtained, b) how you documented participant consent, and c) whether the ethics committees/IRB approved this consent procedure.

3. We note that there is identifying data in the Supporting Information file <CFA data set.xlsx and EFA data set.xlsx>. Due to the inclusion of these potentially identifying data, we have removed this file from your file inventory. Prior to sharing human research participant data, authors should consult with an ethics committee to ensure data are shared in accordance with participant consent and all applicable local laws.

-Location data

Please remove or anonymize all personal information, ensure that the data shared are in accordance with participant consent, and re-upload a fully anonymized data set. Please note that spreadsheet columns with personal information must be removed and not hidden as all hidden columns will appear in the published file.

Additional Editor Comments: Please see the reviewers' comments and make appropriate corrections accordingly.

Reviewers' comments:

Reviewer's Responses to Questions

**Comments to the Author**

1. Is the manuscript technically sound, and do the data support the conclusions?

Reviewer #1: Yes

Reviewer #2: Partly

2. Has the statistical analysis been performed appropriately and rigorously? 

Reviewer #1: Yes

Reviewer #2: Yes

3. Have the authors made all data underlying the findings in their manuscript fully available?

Reviewer #1: Yes

Reviewer #2: Yes

4. Is the manuscript presented in an intelligible fashion and written in standard English?

Reviewer #1: Yes

Reviewer #2: Yes

5. Review Comments to the Author

Reviewer #1: The manuscript describes a coherent study focused on the development and validation of a scale measuring students’ attitudes towards AIGC feedback in English pronunciation learning. The study is ethically appropriate with proper approval and consent processes. There are no obvious issues pertaining to dual publication, research ethics, or publication ethics. The disclosure of funding and competing interests is stated clearly and is acceptable. The methodological EFA, CFA, and concurrent validity analysis of the scale’s rigor, and transparency, strengthens the study’s reliability. The work makes a meaningful impact in the area of AI-assisted language learning, in addition to providing a practical resource for guiding future research and improving instruction.

Reviewer #2: The manuscript presents a well-structured, original study that contributes a validated instrument for measuring students’ attitudes toward AIGC feedback in pronunciation learning. Ethical approval and informed consent were appropriately obtained. There are no concerns about dual publication or research ethics. However, slight overlap in discriminant validity warrants further refinement.

6. PLOS authors have the option to publish the peer review history of their article (what does this mean? ). If published, this will include your full peer review and any attached files.

**Do you want your identity to be public for this peer review?** For information about this choice, including consent withdrawal, please see our Privacy Policy .

Reviewer #1: No

Reviewer #2: No

---

## [Author Response · Author response to Decision Letter 1]

8 Sep 2025

Dear Editor,

We sincerely thank you and the reviewers for the time and effort spent in evaluating our manuscript. We highly appreciate the constructive and insightful comments, which have helped us improve the quality and clarity of the paper. In the revised version, we have carefully addressed all the comments provided by both the academic editor and the reviewers, and we have made corresponding revisions throughout the manuscript. A point-by-point response to each comment is provided below, with explanations of the changes made and the exact locations in the revised manuscript where these changes can be found.

(1) In the Ethical Considerations section, we have added a description of the informed consent procedure. Specifically, the consent was obtained in electronic format: participants were required to read the informed consent form embedded in the online questionnaire, and by clicking the “Agree to participate” option, they indicated their consent to participate. Only after providing consent in this way could they proceed to complete the survey.

(2) To protect participants’ privacy, we have removed the “major” information from the dataset and re-uploaded it as an anonymized file.

(3) We have revised the manuscript to comply with PLOS ONE guidelines regarding Response to Reviewers. Descriptive captions have been added for all files at the end of the manuscript: S1 Table, EFA dataset; S2 Table, CFA dataset; S1 File, the questionnaire. All in-text citations referring to these files have been updated to match the new captions.

(4) We thank the reviewer for pointing out the concern over discriminant validity. We acknowledge that all participants were drawn from the same school, which may have introduced similarities in prior instruction, learning environment, and experience, thereby increasing the perceived overlap between the Personalization and Clarity dimensions. The slightly elevated HTMT value for Personalization likely reflects this sample homogeneity rather than a flaw in the scale design. We have added a cautionary note in the Discussion section and suggested that future studies with more diverse samples be conducted to further validate the distinctiveness of these constructs.

(5) We have carefully checked and revised the manuscript to ensure that it meets all PLOS ONE style requirements, including file naming, formatting of the main text, title, authors, affiliations, and references.

Best regards

---

## [Decision Letter · Decision Letter 1]

8 Oct 2025

Scale for Students’ Attitude towards AIGC Feedback in English Pronunciation Learning: Development, Validation and Application

PONE-D-25-26307R1

**Dear Dr. Yang**

We’re pleased to inform you that your manuscript has been judged scientifically suitable for publication and will be formally accepted for publication once it meets all outstanding technical requirements.

Kind regards,

Jie Zeng, Ph.D. Professor

Academic Editor

PLOS ONE

**Additional Editor Comments (optional):**

**The reviewers have compeleted the second round of reviewing and they reported this manuscript can be accepted with minor language proofreading and revisions. I also recommend accept with finalization.**

**Reviewers' comments:**

**Comments to the Author** 

Reviewer #1: The manuscript makes a timely and meaningful contribution through the development and validation of a scale on student attitude toward AIGC feedback in English pronunciation learning. The study is rigorous and ethical well-designed. Minor worries about discriminant validity and sample diversity are noted and adequately dealt with in the discussion.

Reviewer #2: The manuscript entitled Scale for Students’ Attitude towards AIGC Feedback in English Pronunciation Learning: Development, Validation and Application presents a well-executed and timely study. The work is relevant to current developments in AI-supported language learning and follows a strong methodological foundation. Below is an assessment based on core evaluation criteria:

Soundness of Methodology and Data Support

The research design is methodologically sound. The authors adopt a structured approach to scale development, starting from qualitative item generation, expert validation, pilot testing, and subsequent validation via EFA and CFA. The sample sizes for both phases are adequate and align with accepted standards in psychometric research. The findings clearly support the four-factor structure, which is both theoretically and empirically justified.

Robustness of Statistical Analysis

The statistical analysis is executed with rigor and transparency. The manuscript reports:

Appropriate and well-explained use of EFA and CFA.

Strong model fit indicators (CFI = 0.960, TLI = 0.951, RMSEA = 0.080).

High internal consistency across all dimensions (Cronbach’s alpha > 0.92).

Comprehensive validation procedures, including convergent, discriminant (HTMT), and concurrent validity tests.

One minor limitation was noted: the HTMT score between the Personalisation and Clarity factors (0.943) is slightly above the ideal threshold. However, this is acknowledged and reasonably explained by the authors.

Data Transparency

The authors comply fully with PLOS ONE’s data policy. All data are stated to be freely available in the manuscript and supplementary materials, with no restrictions on access or reuse.

Language Quality and Readability

The manuscript is clearly written in academic English, with accurate terminology and coherent structure. It is easy to follow, though a few minor stylistic improvements (e.g., reducing passive voice, eliminating redundancy) could enhance overall readability. No major language issues are present.

Ethical Compliance

Ethical considerations have been addressed appropriately. IRB approval and informed consent from participants are clearly reported. There are no ethical concerns or issues regarding duplication or plagiarism.

Final Recommendation

This paper offers a valuable and well-founded contribution to the field of AI in language learning. The developed scale is relevant, psychometrically sound, and potentially useful for both research and instructional applications related to pronunciation feedback.

Recommendation: Accept with minor language and editorial revisions.

---

## [Editor Report · Acceptance letter]

PONE-D-25-26307R1

PLOS ONE

Dear Dr. Yang,

I'm pleased to inform you that your manuscript has been deemed suitable for publication in PLOS ONE. Congratulations! Your manuscript is now being handed over to our production team.

Kind regards,

on behalf of

Professor Jie Zeng

Academic Editor

PLOS ONE